# A *Chlamydia pneumoniae* adhesin induces phosphatidylserine exposure on host cells

Jan N. Galle [1], Tim Fechtner[1], Thorsten Eierhoff[2,3,4,5], Winfried Römer [3,4] & Johannes H. Hegemann [1]*

In mammalian cells, the internal and external leaflets of the plasma membrane (PM) possess different phospholipids. Phosphatidylserine (PS) is normally confined to the inner (cytoplasmic) membrane leaflet. Here we report that the adhesin CPn0473 of the human pathogenic bacterium *Chlamydia pneumoniae (Cpn)* binds to the PM of human cells and induces PS externalization but unexpectedly not apoptosis. PS externalization is increased in human cells exposed to infectious *Cpn* cells expressing increased CPn0473 and reduced in exposure to *Cpn* expressing decreased CPn0473. CPn0473 binds specifically to synthetic membranes carrying PS and stimulates pore formation. Asymmetric giant unilamellar vesicles (GUVs) in which PS is restricted to the inner leaflet reveal that CPn0473 induces PS externalization in the absence of other proteins. Thus our identification of CPn0473 as a bacterial PS translocator capable of specific and apoptosis-independent PS externalization during infection extends the spectrum of mechanisms intracellular pathogens use to enter host cells.

[1] Lehrstuhl für Funktionelle Genomforschung der Mikroorganismen, Heinrich-Heine-University Düsseldorf, Universitätsstraße 1, 40225 Düsseldorf, Germany. [2] Lehrstuhl für Molekulare Evolution, Heinrich-Heine-University Düsseldorf, Universitätsstraße 1, 40225 Düsseldorf, Germany. [3] Faculty of Biology, Albert-Ludwigs-University Freiburg, Schänzlestraße 1, 79104 Freiburg, Germany. [4] Signalling Research Centres BIOSS and CIBSS, Albert-Ludwigs-University Freiburg, Schänzlestraße 18, 79104 Freiburg, Germany. [5] Department of Vascular and Endovascular Surgery, University Hospital Münster, Albert Schweitzer Campus 1, 48149 Münster, Germany. *email: johannes.hegemann@hhu.de

The mammalian plasma membrane (PM) exhibits a distinct phospholipid (PL) asymmetry, which is established and maintained by specific ATP-driven lipid translocators (flippases, floppases)[1,2]. Transmembrane movement of phosphatidylserine (PS) from the inner to the outer leaflet of the PM is called externalization, a process executed by scramblases, which occurs in apoptotic cells (triggering their removal by phagocytes) and during platelet activation[3,4]. However, the precise molecular mechanisms that induce or modulate PS externalization are poorly understood.

*Chlamydia pneumoniae* is an important obligate intracellular Gram-negative bacterial pathogen that infects the upper and lower respiratory tract and is implicated in a wide range of chronic diseases including atherosclerosis and Alzheimer's disease[5]. The infectious chlamydial elementary body (EB) gains entry to host cells by receptor-mediated endocytosis and replicates in a membrane-bound compartment called an inclusion[6–9]. We previously identified CPn0473 as a *Cpn*-specific adhesin involved in EB internalization and showed that recombinant CPn0473 (rCPn0473) dose dependently enhances EB entry into host cells, thus promoting infection[10].

In this manuscript, we demonstrate that the chlamydial adhesin CPn0473 binds negatively charged PLs and its C terminus inserts into the host-cell PM. CPn0473 acts as a PL translocator, which transports host-cell PS from the inner cytosolic leaflet of the PM to the extracellular leaflet. CPn0473-mediated PS externalization does not induce apoptosis. CPn0473 binding to synthetic asymmetric membranes results in directional PS translocation from the inner (PS-high) to the outer (PS-low) leaflet. PS translocation via CPn0473 facilitates uptake of the infectious chlamydial EB by the host cell.

## Results

**CPn0473 associates with negatively charged PLs**. To investigate the protein's function further, we screened for interaction partners and found that recombinant rCPn0473 binds strongly to liposomes in pull-down experiments (Fig. 1a). Assays performed with nitrocellulose strips spotted with different lipids indicated that rCPn0473 binds to various negatively charged PLs (Supplementary Fig. 1a). Using giant unilamellar vesicles (GUVs) made of defined lipid mixtures, we confirmed the specificity of rCPn0473 for negatively charged PLs, and verified that rCPn0473 did not bind GUVs consisting of phosphatidylcholine (DOPC). Strikingly, rCPn0473 showed a markedly higher affinity for PS (DOPS) than for phosphatidic acid (DOPA, PA), whereas phosphatidylinositol 4,5-bisphosphate (PI(4,5)P2) was not recognized (Fig. 1b, c). Raising the fraction of PS in the lipid mixture from 5 to 20 mol% enhanced binding of rCPn0473 (Fig. 1d), comparable to the positive control rLactC2, which exclusively binds PS[11] (Supplementary Fig. 1a–c, Fig. 1e). As invasion by *Cpn* EBs occurs via cholesterol-rich membrane domains[10,12], we also tested GUV membranes supplemented with cholesterol, which indeed increased binding to GUVs generally. Interestingly, a rCPn0473 variant lacking the domain required for binding to human cells (BD, aa 307–356)[10] adhered to GUVs, whereas removal of residues 1–171 (essential for stimulating EB internalization[10]) abrogated the interaction. Conversely, fusion of the first 176 aa of CPn0473 to OmcB enabled the latter to adhere to PS-containing GUVs (Fig. 1e, f, Supplementary Fig. 1c).

**Efficient chlamydial infection depends on recognition of PS**. To assess the relevance of PS for infection by *Cpn*, we infected the PS-deficient mutant cell line CHO-PSA3[13]. We observed a significant reduction in *Cpn* infection relative to wild-type CHO-K1 cells, indicating that PS is important for infection (Fig. 2a).

Next, we tested whether the rCPn0473-mediated boost in EB internalization depends on host-cell PS[10]. CHO-K1 cells pre-exposed to rCPn0473 indeed exhibited an increase in internalized EBs compared with bovine albumin (BSA)-treated cells, but no effect was observed in the PS-deficient line pre-treated with rCPn0473 (Fig. 2b). Importantly, no significant difference in rCPn0473 binding was noted between K1 and PSA3 cells (Supplementary Fig. 1d). Overall, these data indicate that CPn0473 binds to PS, and that PS is essential for the rCPn0473-mediated boost in both EB internalization and infectivity.

Interestingly PS, which under homeostatic conditions is restricted to the inner leaflet of the PM[14], is externalized upon infection with *Cpn*[15]. Indeed, when we assayed *Cpn*-infected cells for PS externalization with fluorescent annexin-V[16], a signal was observed surrounding adhering EBs as early as 5 min post infection (pi), whereas heat-inactivated *Cpn* EBs did not induce fluorescence (Fig. 2c). At 60 min pi, $83 \pm 9\%$ of *Cpn* EBs were annexin-V-positive ($\pm$ = standard deviation (SD) of mean data, Fig. 2d). Interestingly, the association of adhering *Cpn* EBs with annexin-V signals was species-specific, as only $15 \pm 3\%$ of *C. trachomatis* EBs of serovar E and only $9 \pm 1\%$ of serovar LGV exhibited annexin-V fluorescence (Fig. 2d). To confirm this result, we assayed PS exposure by *Cpn* using rLactC2, and again detected externalized PS at EB entry sites. Interestingly, both *Cpn* and rLactC2 signals accumulated in cholera toxin (CTxB)-positive, cholesterol-enriched membrane domains[17] (Supplementary Fig. 2a).

**CPn0473 induces PS externalization**. As CPn0473 is specific to *Cpn*, and only *Cpn* EBs showed massive PS externalization, we tested whether rCPn0473 itself could induce PS translocation by incubating HEp-2 cells with rCPn0473. Indeed, we observed PS externalization at the binding site of rCPn0473 in a time- and dose-dependent manner visualized by Annexin-V and rLactC2-fluorescence intensity (FITC), whereas a triple-AAA mutant version of rLactC2-FITC that does not bind PS failed to stain rCPn0473-bound cells (Fig. 3a–c, Supplementary Fig. 2b). Importantly, when using the FITC-labeled PI(4,5)P2-binding domain of PLCγ and the PI(3)P-binding FYVE domain of the Hgs (Supplementary Fig. 1a, e, f), we did not detect any externalization of phosphatidylinositol 4,5-bisphosphate (PI(4,5)P2) and phosphatidylinositol 3-phosphate (PI(3)P), two other negatively charged PLs that are normally restricted to the inner leaflet of the PM, to the host-cell surface upon the addition of rCPn0473 to HEp-2 cells (Supplementary Fig. 2c), evidencing specificity of rCPn0473 for PS. The extent of annexin fluorescence correlated with the amount of rCPn0473 used, and increased over time, first becoming detectable after 15 min of incubation (Fig. 3b, c, Supplementary Fig. 3a). As calcium entry might trigger cellular PS exposure, we tested different calcium concentrations in the medium of HEp-2 cells. Notably, the level of rCPn0473-induced PS externalization was not enhanced by elevated extracellular calcium levels (Fig. 3b).

**PS externalization mediated by CPn0473 is non-apoptotic**. Further characterization of the CPn0473-induced PS externalization process revealed that it is sensitive to cholesterol depletion by methyl-beta-cyclodextrin (MβCD) and did not rely on an intact cortical microtubule and actin meshwork as proofed by nocodazole or cytochalasin D treatment (Fig. 3d, Supplementary Fig. 3b–d). PS externalization is a hallmark for apoptosis, which is triggered by the proteolytic activation of caspase-3 and inactivation of PARP. Importantly, no cleavage of pro-caspase-3 or PARP could be detected in cells exposed to rCPn0473 or in *Cpn*-infected cells, in contrast to staurosporine-treated, apoptotic cells (Fig. 3e).

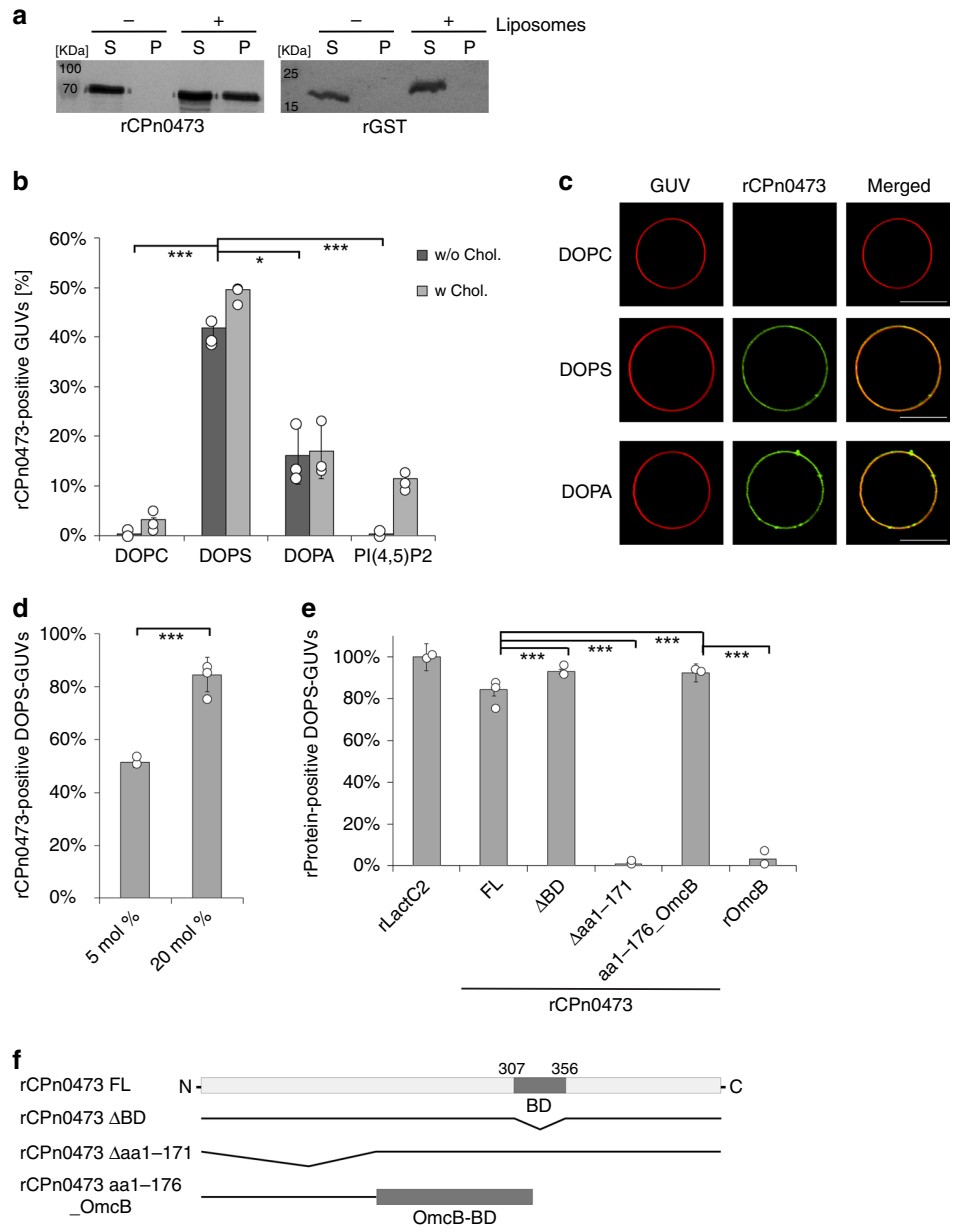

**Fig. 1** Recombinant CPn0473 interacts preferentially with phosphatidylserine (PS). **a** Full-length rCPn0473 binds to liposomes (L-α-phosphatidylcholine, phosphatidylethanolamine, carbohydrates and other lipids). Pellet (P) and supernatant (S) fractions were analyzed on Western blots ($n = 3$). **b**, **c** Binding efficiency of rCPno473 FL to giant unilamellar vesicles (GUVs). GUVs consisting of phosphatidylcholine (DOPC) plus the indicated lipids were incubated with DyLight488-labeled proteins (DOPS: phosphatidylserine, DOPA: phosphatidic acid, PI(4,5)P2: phosphatidylinositol 4,5-bisphosphate). ($\geq 75$ GUVs were examined per experiment, $n = 3$). Mean ± standard deviation (s.d.). scale bar: 10 μm. **d** Effect of PS concentration on rCPn0473 binding. Mean ± s.d. ($n = 2$). **e** Binding of rLactC2 ($\geq 60$ GUVs examined per experiment, $n = 3$), rCPn0473 deletion variants ($\geq 50$ GUVs examined per experiment, $n = 2$) and OmcB derivatives to GUVs. FL: full-length, BD: host-cell binding domain, aa: amino acid. Mean ± s.d. **f** CPn0473 variants used in **e**. For all panels: ***$P <$ 0.001, **$P < 0.01$ and *$P < 0.05$, n.s. not significant ($P > 0.05$) (Student's two-sample $t$ test). Source data are provided as a Source Data file

Importantly, exposure of HEp-2 cells to the chlamydial adhesins rPmp21 or rOmcB did not induce PS externalization (Supplementary Fig. 4a).

Next, we modulated the amount of CPn0473 on the surface of infectious EBs prior to infection of HEp-2 cells. Coating of EBs with rCPn0473 resulted in EB-associated PS signals that were more than three times as intense as those in untreated EBs, and correlated well with the total amount of CPn0473 associated with EBs (Fig. 3f, Supplementary Fig. 4b). Correspondingly, blocking endogenous CPn0473 accessible on the EB surface by pretreatment with anti-CPn0473 antibodies reduced PS externalization relative to untreated EBs (Fig. 3g, Supplementary Fig. 4b).

Exposure to an anti-Pmp21 antibody had little effect on PS externalization.

To identify the region of CPn0473 responsible for PS externalization, we first tested the variant lacking the BD domain (Supplementary Fig. 1d)[10]. rCPn0473ΔBD is unable to bind HEp-2 cells[10] and, as expected failed to induce PS externalization. Likewise, the deletion variant lacking the first 171 aa, which is incapable of PS-binding, was also unable to externalize PS. However, a variant lacking aa 150–255 was still able to externalize PS, indicating that the N-terminal aa 1–149 are essential for PS externalization (Supplementary Fig. 4c).

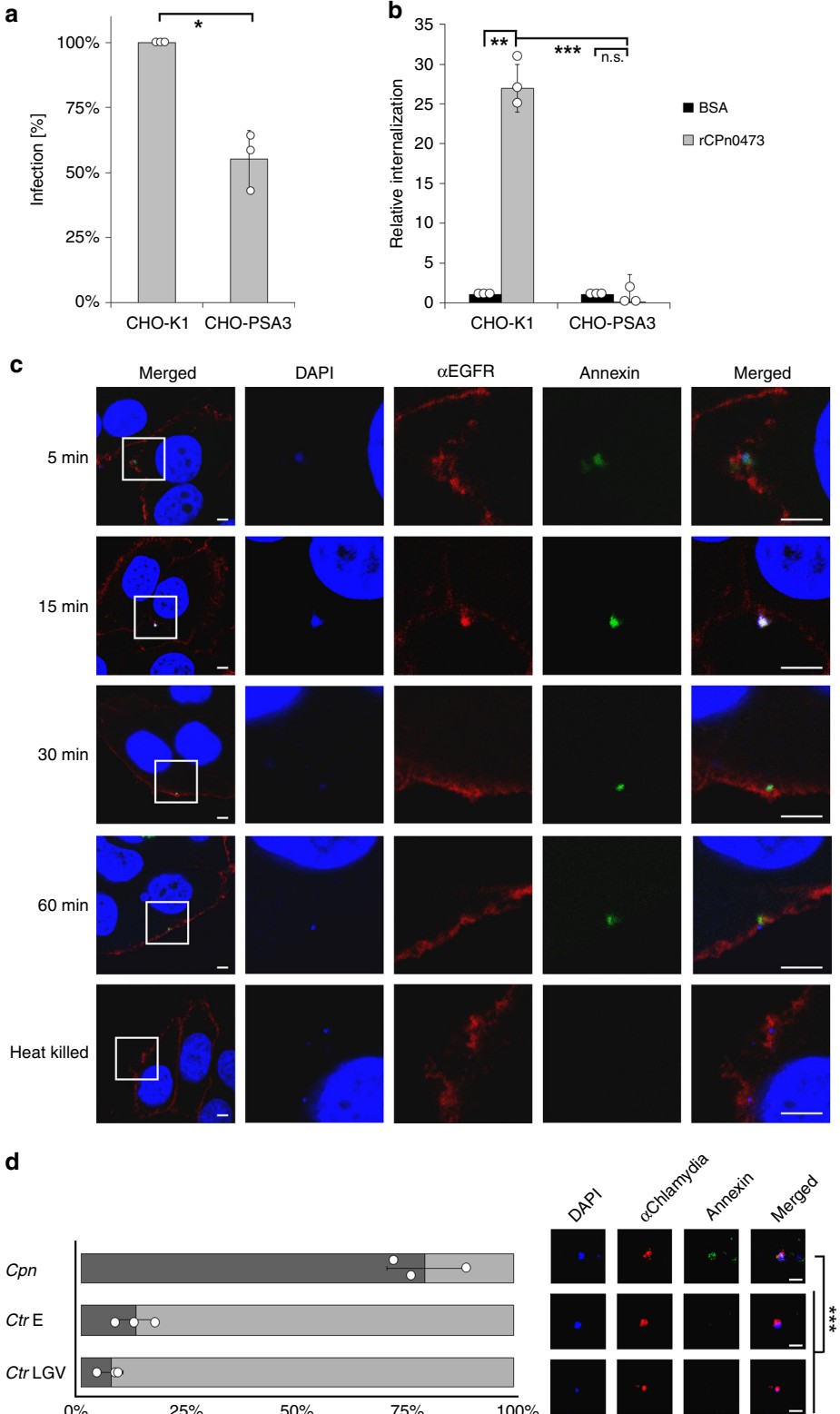

**Fig. 2** *C. pneumoniae* specifically induces externalization of phosphatidylserine (PS). **a** Infectivity of *C. pneumoniae* EBs (MOI = 10) in wild-type (WT) and PS-deficient (PSA3) CHO cells (*n* = 3). mean ± s.d. Infectivity in WT CHO cells was set to 100%. Mean ± s.d. **b** qRT PCR analysis of internalized *Cpn*. Triplicate samples of EBs in CHO WT and PSA3 cells (2 hpi) were pre-treated with rCPn0473 (100 μg/ml) for 1 h prior to infection (MOI = 10) (*n* = 3). Mean ± s.d. **c** Externalization of PS by *C. pneumoniae* EBs early in infection. HEp-2 cells were infected with chlamydial EBs for the indicated times (MOI = 10). Externalized PS was stained with annexin-V-FLUOS prior to fixation, followed by staining with DAPI and anti-EGFR antibody. Scale bars: 2.5 μm. **d** Externalization of PS by the indicated chlamydial species. HEp-2 cells were infected with chlamydial EBs for 1 h (MOI = 10). Externalized PS was stained as in **c**. Mean (triplicates) ± s.d. (*n* = 3). For all panels: ***$P < 0.001$, **$P < 0.01$ and *$P < 0.05$, n.s. not significant ($P > 0.05$) (Student's two-sample *t* test). Source data are provided as a Source Data file

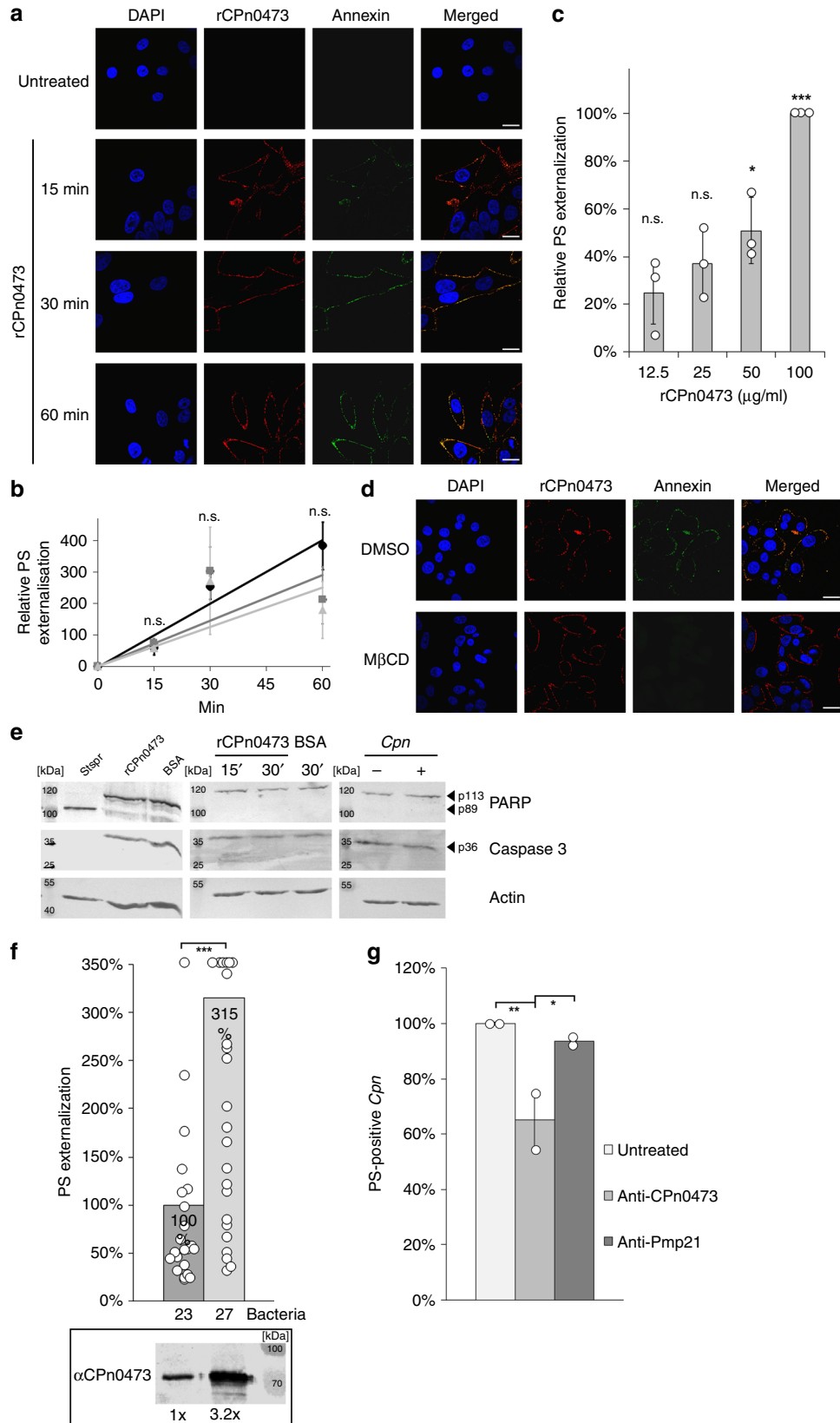

**CPn0473 inserts into host membranes via its C terminus**. The data presented thus far are compatible with the hypothesis that CPn0473 itself is a PS translocator. To elucidate the membrane orientation of rCPn0473 during binding to the host-cell PM, His-tagged rCPn0473 was site-specifically biotinylated at single cysteine (Cys) added to the N- or C terminus, and incubated with HEp-2 cells, which were then exposed to β-mercaptoethanol. If the biotinylated terminal Cys is accessible to reduction by β-mercaptoethanol, the biotin moiety should be lost. Indeed, after treatment of cells with the N-terminally biotinylated CPn0473,

**Fig. 3** Recombinant CPn0473 induces externalization of phosphatidylserine (PS). Externalized PS was stained using annexin prior fixation and antibody staining **a–f**. **a–c** Fluorescence analysis reveals time- and concentration-dependent PS externalization by rCPn0473. Representative images are shown in **a**, scale bars: 10 μm. **b** Fluorescence analysis of externalized PS on HEp-2 cells, incubated with recombinant CPn0473 (100 μg/ml) in DMEM media with different calcium levels (circle (1.8 mM), square (2.8 mM), triangle (4.8 mM)). Mean ± s.d. ($n = 3$). **c** Concentration dependence of PS externalization by rCPn0473. Mean ± s.d. ($n = 3$). **d** Cholesterol depletion with 5 mM MβCD reduces the efficiency of rCPn0473-induced PS externalization. Scale bars: 10 μm. **e** Western blot analysis of apoptosis marker proteins (caspase-3 and PARP). Apoptosis was induced by incubation with 2.5 μM staurosporine (Stspr). Arrowheads mark the inactive procaspase-3 (p36), full-length PARP (p113), and cleaved PARP (p89) ($n = 3$). **f** Coating of *C. pneumoniae* EBs with 1 μM rCPn0473 at 1 hpi (MOI = 10) boosts levels of PS externalization. CPn0473 levels were determined by Western blotting. The mean of one typical experiment is shown ($n = 3$). **g** Masking of rCpn0473 by pre-incubation with specific antibodies reduces PS externalization. *C. pneumoniae* EBs were pre-incubated with CPn0473-specific antibodies (diluted 1:100) at MOI = 20. At least 70 DAPI- and CPn0473-positive EBs analyzed per experiment. Mean ± s.d. ($n = 2$). For all panels: \*\*\*$P < 0.001$, \*\*$P < 0.01$, and \*$P < 0.05$, n.s. not significant ($P > 0.05$) (Student's two-sample *t* test). Source data are provided as a Source Data file

only 6 ± 5% of the protein remained biotinylated, whereas 39% (±1%) of the C-terminally modified form survived exposure to the reductant, indicating that the C terminus may be inserted into the host PM (Fig. 4a). Indeed, a transmembrane region is predicted near the C terminus (TM, aa 396–425) by ProtScale Hphob[18] (Fig. 4b). To determine whether rCPn0473 binding alters membrane integrity, we monitored the stability of carboxyfluorescein (CF)-loaded DOPS-GUVs in the presence of Alexa594-labeled full-length (FL) rCPn0473 or a TM deletion variant (ΔTM). Upon incubation with rCPn0473 FL, the CF signal within the GUV lumen decreased within 5 min, and this was followed by an increase in the luminal Alexa594 signal. Although a 50% loss of CF (MW: 376.32 Da) was observed within ~2.5 min, it took 8.5 min to reach 50% of the maximal Alexa594-rCPn0473 (MW: 55 kDa) uptake into the GUV lumen. In contrast, binding of the ΔTM variant to PS-containing GUVs did not alter luminal CF fluorescence nor result in increased luminal Alexa594 fluorescence (Fig. 4c–e). Thus, rCPn0473 binding to DOPS-GUVs makes the membrane permeable to small and large molecules without affecting overall GUV stability.

**CPn0473 is a chlamydial PS translocator**. In order to test directly for PS translocator activity of rCPn0473, we prepared GUVs bearing PS predominantly in one of the membrane leaflets[19,20]. PS asymmetry was verified using FITC-rLactC2. Mixing GUVs carrying PS in the outer leaflet (PS_{out}-GUV) with rLactC2 immediately resulted in strong FITC fluorescence at time point 0 min, which declined over time. GUVs carrying PS localized mainly in the inner leaflet (PS_{inn}-GUV) showed little FITC-rLactC2 fluorescence under the same conditions (Fig. 4f, g). However, when PS_{inn}-GUVs were preloaded with FITC-rLactC2, the luminal rLactC2 stained inner leaflet PS, resulting in significant fluorescence and confirming PS asymmetry (Fig. 4f, Supplementary Fig. 4d, e).

Crucially, simultaneous exposure of PS_{inn}-GUVs to external rCPn0473 and FITC-rLactC2 immediately resulted in FITC-rLactC2 fluorescence, which increased over the next 30 min (Fig. 4f, g). On the other hand, when PS_{out}-GUVs, prepacked with FITC-rLactC2, were exposed to external rCPn0473, we observed no FITC-rLact2 fluorescence at the inner leaflet (Supplementary Fig. 4d, e). These results argue against a scrambling mechanism for rCPn0473, and thus support directional rCPn0473-mediated PS translocation from the inner (PS-high) to the outer (PS-low) leaflet in a synthetic membrane system.

## Discussion

Our data identify the *Cpn* adhesin CPn0473 as a lipid translocator that mediates PS externalization at the host–cell PM surrounding the adhering pathogen. We therefore refer to the protein hereafter as LIPP (for 'Lipid-dependent Internalization Promoting

Protein'). LIPP is found on the surface of adhering *Cpn* EBs[10]. Our model implies that LIPP interacts via its BD domain with the PM via an as yet unknown cell surface structure associated with cholesterol-rich microdomains. The LIPP–PM interaction likely triggers both conformational changes within the protein and oligomerization. Together, these changes enable the N-terminal domain of LIPP to interact with the PM and translocate PS from the inner to the outer leaflet of the bilayer—which in turn enhances EB internalization. This scenario could explain why the assembly and the activity of the LIPP translocation complex acts rather slowly by comparison with known integral membrane flippases and scramblases. Notably, LIPP activity requires neither ATP nor $Ca^{2+}$, although a potentially beneficial effect of intracellular ATP- or $Ca^{2+}$-stores cannot be ruled out at the moment. LIPP is therefore the first known lipid translocator protein that can gain access to the inner leaflet of the PM from the extracellular side.

How might exofacial PS localization support EB internalization? *Cpn* is an obligate intracellular pathogen, and EB internalization critical for infection. Thus *Cpn* may have invented PL translocase activity as a mechanism to expose a specific host-cell ligand, i.e., PS, to which the LIPP adhesin can then bind. Surface-exposed PS also acts as a ligand during apoptotic clearance and blood cell clotting. Moreover, PS externalization promotes HIV uptake[21]. However, in these latter cases, PS exposure occurs via activated human scramblases. Exofacial PS has also been found to have a role during host-cell entry by other bacterial pathogens, but the underlying mechanisms are not understood[21–23].

Translocation of PS will also modulate PM structure. Under homeostatic conditions, the anionic PS is restricted to the inner leaflet of the PM and its head-group experiences electrostatic repulsion, which is limited by cholesterol (also enriched in the inner leaflet)[24,25]. Specific PS translocation around the adhering EB increases the anionic charge density of the exofacial leaflet which, in the absence of sufficient cholesterol, will develop outward curvature at the edges of the invaginating membrane owing to electrostatic repulsion, thus supporting EB endocytosis. This process might be facilitated by host scaffolding proteins, which are known to enhance membrane curvature and thereby vesicle formation[26]. Finally, transient PS depletion from the inner PM leaflet surrounding the adhering *Cpn* EBs will result in a relocalization of specific PS-binding proteins normally found associated with the inner bilayer leaflet, which would otherwise impinge on the entry process. Hence, our findings not only contribute to our knowledge of basic host-pathogen interactions, they suggest that inhibiting PS exposure and/or pathogen–PS interaction might open a route to therapies for bacterial and viral diseases.

## Methods

**Chemicals and antibodies**. MβCD, nocodazole, Cell Dissociation Solution Non-enzymatic 1×, AP-coupled streptavidin, CF and FITC-conjugated cholera toxin B

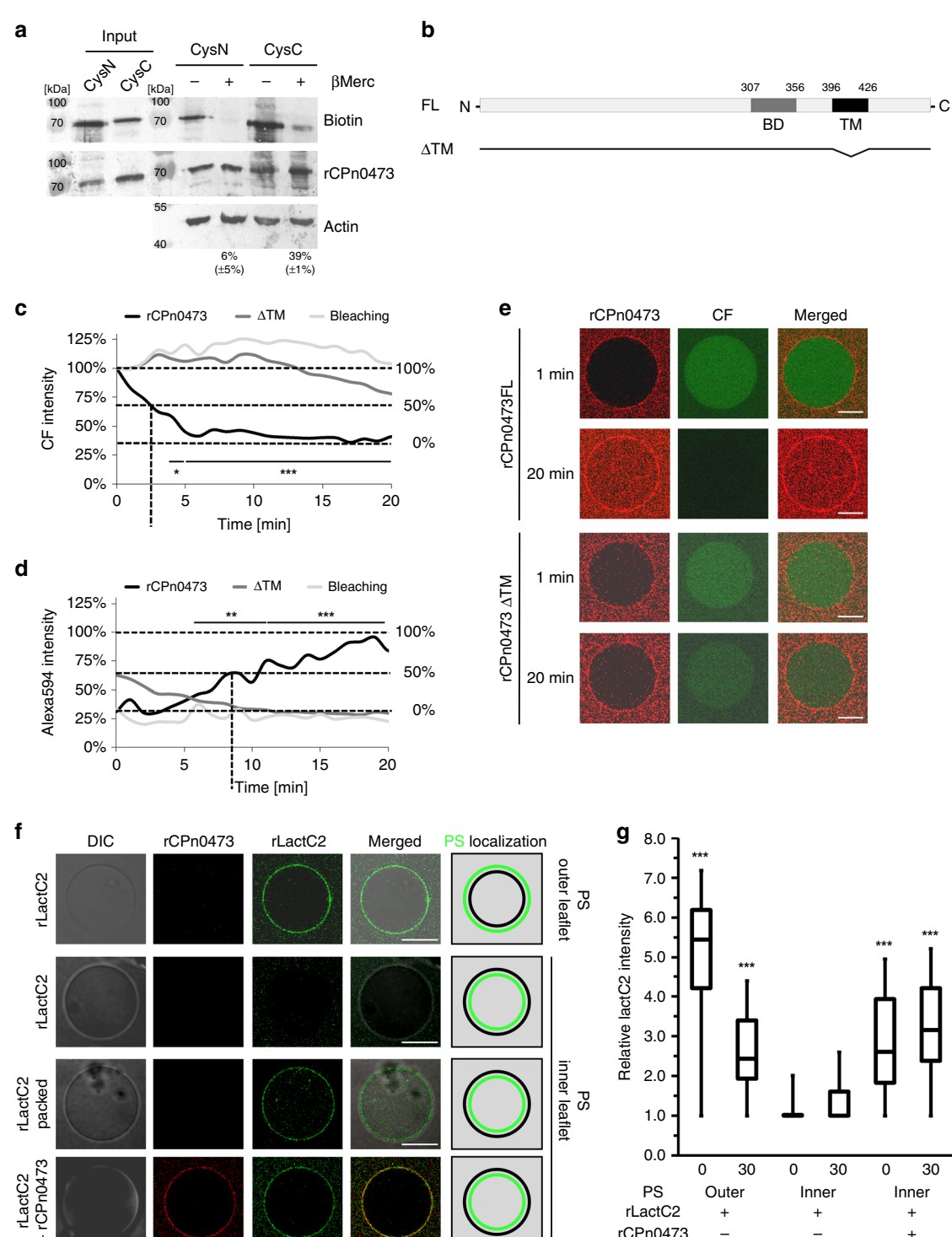

subunit (FITC-CTxB) were obtained from Sigma-Aldrich, carboxyfluoresceindiacetate-succinimidylester (CFSE) and the lipids DOPS, DOPC, PI(3)P, PI(4,5)P$_2$, DOPA, TR-DHPE from Life Technologies, gentamicin was sourced from Invitrogen, proteinase K from Millipore and the SYBR Green PCR Master Mix from Applied Biosystems. Staurosporine, anti-caspase-3 antibodies (1:100, Western blot (WB)), anti-PARP antibodies (1:100, WB), and annexin-V were kindly provided by Christoph Peter (Institute for Molecular Medicine, HHU, Düsseldorf). Annexin was also obtained directly from Roche. EZ-Link HPDP-biotin, cytochalasin D and rhodamine/phalloidin was obtained from Thermo Scientific, the lipid membrane strips were from Echelon Biosciences. Commercially available antibodies were obtained from the following sources: anti-β-actin (#A1978, 1:2000 (WB)) from Sigma-Aldrich; Alexa488- and Alexa594-conjugated anti-rabbit and anti-mouse antibodies from Life Technologies (all 1:200 (Immunofluorescence (IF)); anti-GST (#C2309, 1:1000 (WB)) from Santa Cruz; anti-PentaHis (#34660, 1:2500 (WB)) from Qiagen; AP-conjugated anti-mouse (#S372, 1:7500 (WB)) and anti-rabbit (#S3731, 1:7500 (WB)) from Promega; anti-LPS (Pathfinder) (#30701, 1:4 (IF)) from BioRad, anti-tubulin from Acris, anti-EGFR (# MA5-13269, 1:50 (EGFR)) from Invitrogen. The anti-Pmp21 (1:50 (WB); 1:10 (IF)) and anti-CPn0473 (1:25 (WB); 1:5 (IF)) antibodies were produced in rabbit by Eurogentec, and anti-MOMP (1:100 (WB)) was generated by G. Zhong (University of Texas Health Science Center at San Antonio). Phosphate-buffered saline (PBS) buffer used in the experiments consists of 137 mM NaCl, 2.7 mM KCl, 10 mM Na$_2$HPO$_4$, 1,8 mM KH$_2$PO$_4$. Hank's Buffered Saline Solution (HBSS) (Thermo Scientific, #14175-053) used in this work consists of 5.3 mM KCl, 0.4 mM KH$_2$PO$_4$, 4.2 mM NaHCO$_3$, 137.9 mM NaCl, 0.3 mM Na$_2$HPO$_4$, 5.6 mM D-Glucose.

**Fig. 4** rCPn0473 acts as a chlamydial phosphatidylserine translocase. **a** Orientation of rCPn0473 in the host-cell membrane. N- or C-terminal single-cysteine variants of Cpn0473 were biotinylated prior to incubation with HEp-2 cells. Accessible (i.e., extracellular) biotin attached to the terminal Cys residue was removed by treatment with β-mercaptoethanol and the amount that remained linked to the protein was analyzed on Western blots using streptavidin. Mean residual biotin is given underneath the Western blot ($n = 3$). **b** Schematic of a CPn0473 deletion variant lacking the predicted transmembrane domain (ΔTM) predicted by ProtScale Hphob. BD: host-cell binding domain. **c–e** Binding of rCpn0473 affects the integrity of PS-containing GUVs. DOPS-GUVs were preloaded with carboxyfluorescein (CF, 1:100 dilution) and incubated with Alexa594-coupled rCPn0473, rCPn0473ΔTM or untreated (Bleaching). Fluorescence intensity of CF **c** and Alexa594 **d** in the GUV lumen was analyzed (at least 15 GUVs) ($n = 2$). The horizontal dotted lines show the differences in signal intensity (0%, 50% and 100% signal intensity) during the course of the experiment. The dotted vertical line represents the time point when a 50% shift in signal intensity is reached. Representative images are shown in **e**. Scale bar: 10 μm. Significant differences between full-length rCPn0473 and rCPn0473ΔTM are indicated. **f**, **g** Asymmetric GUVs were prepared based on[19]. In the presence or absence of rCPn0473, FITC-labeled, recombinant rLactC2, as a marker for PS, was incubated with GUVs bearing PS predominantly in the outer (PS$_{out}$-GUV) or inner (PS$_{inn}$-GUV) leaflet. Representative views are shown in **f**. Scale bar: 10 μm. rLactC2 binding was quantified using ImageJ software. Background intensity was set to 1 (up to 24 GUVs analyzed) ($n = 3$). **g**. The data are represented as a box and whisker plot. The center line represents the mean score, the upper and lower quartiles contain the mean 50% of the data while the whiskers show the minima and maxima of all data. Significance of differences is determined between individual time points (0 min or 30 min). For all panels: ***$P < 0.001$, **$P < 0.01$, and *$P < 0.05$ (Student's two-sample $t$ test). Source data are provided as a Source Data file

**DNA manipulations and protein expression.** *Escherichia coli* XL$_1$ blue (Stratagene) was used for plasmid amplification and the Origami strain (Novagen) for protein expression. Plasmids were constructed by in vivo homologous recombination in *Saccharomyces cerevisiae* strain YKM2, a GFP-expressing version of EBY100 (Invitrogen). All coding sequences were amplified via PCR from chlamydial genomic DNA or cDNA derived from human genomic DNA with a 40 base pair long homologous region on either side for cloning into the respective destination vector in *S. cerevisiae*[6] (Supplementary Table 1). The binding domain of OmcB (aa 40–100) from *C. pneumoniae* was expressed with a C-terminal His-tag and fused N-terminally to GST. GST, Pmp21, CPn0473 FL and the deletion variants were expressed as C-terminally His-tagged proteins. The PS-binding domain of lactadherin (expression clone kindly provided by Dr. S. Grinstein, University of Toronto, Toronto, Canada[16] (Addgene Plasmid # 15247)) was C terminally tagged with GST. The PI(4,5)P$_2$-binding domain of phosphoinositide-phospholipase C gamma (PLCγ) and the PI(3)P-binding FYVE domain of the Hepatocyte growth factor-regulated tyrosine kinase substrate (Hgs/Hrs) were C-terminally tagged with GST (kindly provided by Dominik Spona, HHU, Düsseldorf). GST- and His-tagged proteins were purified using the protocols supplied by Sigma-Aldrich and Qiagen, respectively, and analyzed on Western blots.

**Cultivation of epithelial cells.** Human HEp-2 cells (epithelial larynx carcinoma, ATCC No.: CCL-23) were cultivated in Dulbecco's Modified Eagle Medium (DMEM) GlutaMax with fetal calf serum (FCS), vitamins, non-essential amino acids, amphotericin B (2.5 μg/ml) and gentamicin (50 μg/ml) at 37 °C in an atmosphere containing 6% CO$_2$. CHO cells (Chinese hamster ovary, ATCC No.: CCL. 61) were cultivated in F12K medium supplemented with FCS, vitamins, non-essential amino acids, amphotericin B (2.5 μg/ml) and gentamicin (50 μg/ml) at 37 °C in an atmosphere containing 6% CO$_2$. Wild-type CHO-K1 and the PS-deficient CHO cell line PSA3 (defective in the *pssA* gene encoding PS synthase I activity) were kindly provided by Dr. O. Kuge, Kyushu University, Fukuoka, Japan[13]. The CHO-PSA3 was cultivated using F12K medium containing PS liposomes (30 μM).

**Propagation of chlamydial strains.** The HEp-2 cell line was used for propagation of chlamydial strains (patient isolates). The cells were cultured in DMEM Gluta-Max with FCS, vitamins, non-essential amino acids, amphotericin B (2.5 μg/ml) and gentamicin (50 μg/ml) at 37 °C in an atmosphere containing 6% CO$_2$. *C. pneumoniae* GiD and the *C. trachomatis* serovars L2 (L2/434/Bu) and E (DK-20) were propagated in HEp-2 cells in the presence of 1.2 μg/ml cycloheximide, and EBs were purified using 30% gastrografin (Schering). The CHO cell lines were used for adhesion and infection experiments.

**Immunofluorescence microscopy.** Microscopy was performed with a Nikon Eclipse Ti-E C2 confocal microscope (DS-Qi1MC Camera). Confocal images were assembled using NIS Element software (Nikon). Infected cells were fixed either with 96% methanol for 5 min at room temperature or with 3% para-formaldehyde for 20 min at 4 °C. Where necessary, cells were permeabilized with 2% saponin prior to staining with an antibody or marker protein. In this case, 0.5% saponin was present during the whole staining process. Unspecific binding sites were blocked with 2% BSA and antibodies were dissolved in a 0.05% Tween-20 solution.

**Adhesion assay.** Adhesion assays were carried out with fully confluent layers of $10^6$ HEp-2 cells or CHO cells grown at 37 °C. Binding of soluble recombinant proteins was assessed by overlaying confluent cells with 250 μl of culture medium containing the soluble recombinant protein of interest (100 μg/ml) at 4 °C on ice for 15, 30, or 60 min. After extensive washing, cells were detached, pelleted (5 min

at $300 \times g$) and resuspended in HBSS (100 μl). Bound recombinant protein was then quantified by immunoblotting using anti-His antibodies.

**Membrane localization assay.** To assay the orientation of rCPn0473 FL when bound to the PM of host cells, the C-terminally His-tagged protein was modified by adding a single cysteine (Cys) at either the N- or the C terminus. The two variants were then biotinylated at the terminal Cys residue in the presence of 1 mM EDTA using 0.05 μg of EZ-Link HPDP-Biotin per μg protein for 4 h on ice. Each of the two biotinylated forms (100 μg/ml) were incubated with confluent HEp-2 cells at 37 °C for 30 min. Cells were cooled on ice and incubated with 100 mM β-mercaptoethanol (10 min) to remove extracellular biotin. After extensive washing, cells were detached, using Cell Dissociation Solution, pelleted (5 min at $300 \times g$) and resuspended in HBSS (100 μl). Bound recombinant protein was then quantified by immunoblotting using anti-His antibodies. The amount of terminally Cys-bound biotin remaining was measured using AP-coupled streptavidin and quantified relative to the anti-His signal and the loading control using ImageJ.

**Infection assays using a PS-deficient cell line.** Wild-type CHO-K1 and the PS-deficient CHO cell line PSA3[13] used in infection assays were starved for 2 days in F12K medium lacking PS liposomes (30 μM). Prior to infection with EBs (MOI = 40) at 2 hpi, the medium was replaced by F12K medium containing PS liposomes (30 μM), prepared via the freeze–thaw-sonication method (explained below), and cycloheximide (1.2 μg/ml). At 50 hpi, cells were washed with HBSS and fixed with methanol. Inclusions were stained with Pathfinder. Infectivity was quantified by the standard microscopy assay by counting the numbers of inclusions in three large images (size of 3 × 3 single pictures, 60-fold magnification) (triplicates).

To quantify the effects of recombinant proteins on EB internalization, confluent CHO cells were overlaid with 250 μl of cell culture medium containing the relevant recombinant protein (100 μg/ml), incubated for 1 h, and cooled to 4 °C. Purified EBs were added (at MOI = 40) and incubated for 1 h at 4 °C, then for 2 h at 37 °C to allow internalization. After removal of unbound EBs by extensive washing with HBSS, CHO cells and non-internalized EBs were detached with 0.5 × Trypsin/EDTA solution (Invitrogen) for 15 min at 37 °C. EBs were removed by sequential centrifugation at $1000 \times g$ and $250 \times g$. Internalization was quantified by qPCR using a modified version of a previously described protocol[27]. DNA was isolated from infected cells by phenol-chloroform extraction and qPCR was performed with 200 ng of RNase-treated DNA using the SYBR Green PCR Master Mix (Applied Biosystems) and 16 S rRNA primers (forward: CCAACACCTCACGGCACGAG, reverse: CGCCTGAGGAGTACACTCGC) to detect chlamydial DNA and GAPDH primers (forward: TGGCTACAGCAACAGAGTGG, reverse: GTGAGGGAGATGATCGGTGT) to quantify host DNA. The reaction sequence for qPCR consisted of 5 min at 95 °C, followed by 60 cycles of 95 °C for 15 s, 60 °C for 15 s and 68 °C for 30 s (ABI Prism 7000, Applied Biosystems). The number of 16 S gene copies relative to GAPDH was expressed as $2^{-\Delta\Delta CT}$[27].

**Membrane lipid strip assay.** PL-binding assays were carried out using nitro-cellulose membrane strips bearing (100 pmol) samples of 15 different biologically active lipids (Echelon Biosciences Inc.), following the protocol provided by the supplier. The lipids tested were glyceryl tripalmitate (GT), diacylglycerol (DAG), PA, PS, phosphatidylethanolamine (PE), phosphatidylcholine (PC), phosphatidylglycerol (PG), cardiolipin (CL), phosphatidylinositol (PI), phosphatidylinositol-(4)-monophosphate, (PI(4)P, phosphatidylinositol-(4,5)-diphosphate (PI(4,5)P2), phosphatidylinositol-(3,4,5)-triphosphate (PI(3,4,5)P3), cholesterol (Chol), sphingomyelin (SM), and 3-sulphogalactosyl ceramide (Sulfatide). The nitrocellulose membranes were blocked with PBS-T (0.1% v/v Tween) containing 3% (wt/vol) fatty acid-free BSA for 1 h at room temperature (RT), following 1 h of incubation with the recombinant proteins (1 μg/ml) in PBS-T (+3% BSA). The membranes

were then washed three times with PBS-T, and bound recombinant protein was detected using anti-His or anti-GST antibodies.

**Liposome pull-down experiment**. Liposomes made of 55% L-α-DOPC, 20% phosphatidylethanolamine, carbohydrates and other lipids (Sigma-Aldrich, Type IV-S, #P-3644) were prepared via the freeze–thaw–sonification method (generous gift from J. Wiese, Heinrich-Heine University Düsseldorf). Lipids were resuspended in chloroform and acetone and stirred for 2 h at RT. After precipitation of the PLs overnight at 4 °C, lipids were dissolved in diethyl ether and dried under nitrogen. The PL-pellets were dissolved in HBSS and sonicated in a water bath (30 × 5 s). They were purified by passage over a PD10 column and incubated with recombinant protein (5 μg) for 20 min at room temperature in a final volume of 100 μl. Liposomes were pelleted for 15 min at $16000 \times g$, the supernatant was collected and the pellet was resuspended in 100 μl PBS. Aliquots of the pellet and supernatant fractions were then analyzed by immunoblotting. Recombinant proteins were detected using anti-His antibodies.

**GUV-binding assay**. GUVs were prepared via electroformation as essentially described[28,29]. Lipid-coated ITO slides were dried under vacuum overnight. The following lipid mixtures (in mol%) were spread on two ITO slides [the fluorescent lipid Texas Red-DHPE (Life Technologies) was added for imaging purposes]: DOPC/Texas Red-DHPE: 99.75/0.25, DOPC/TR-DHPE/cholesterol: 74.75/0.25/25, 94.75/0.25/5 DOPC/TR-DHPE/DOPS, 74.75/0.25/5/20 DOPC/TR-DHPE/DOPS/ cholesterol, 59.75/0.25/20/20, DOPC/TR-DHPE/DOPS/cholesterol, 94.75/0.25/5 DOPC/TR-DHPE/DOPA, 74.75/0.25/5/20 DOPC/TR-DHPE/DOPA/cholesterol, DOPC/TR-DHPE/cholesterol, 94.75/0.25/5 DOPC/TR-DHPE/PI(4,5)P$_2$, 74.75/0.25/5/20 DOPC/TR-DHPE/PI(4,5)P$_2$/cholesterol, 69.75/0.25/10/20 DOPC/ TR-DHPE/PI(4,5)P$_2$/cholesterol or 69.75/0.25/10/20 DOPC/TR-DHPE/PI(3)P/ cholesterol. The two ITO slides were combined with the space between filled with sucrose (10%, w/v) and incubated at RT for 1.5 h (electroformation). The GUVs were added at room temperature to DyLight488-labeled rCPn0473 (100 μg/ml) and immediately observed on a confocal fluorescence microscope (Nikon Eclipse Ti-E with A1R confocal laser scanner, 60x oil objective, NA = 1.49). Image acquisition and analysis was performed with NIS-Elements (Nikon).

**GUV lipid translocator assay**. Asymmetric GUVs were prepared as essentially described previously[19]. The following lipid mixtures (in mol%) were used for the preparation of GUVs: 60/20/20, DOPC/DOPS/cholesterol and 80/20, DOPC/cholesterol. A total of 115 μg lipid mass per membrane leaflet was dried and resuspended in mineral oil (200 μl per leaflet). The outer leaflet was prepared by overlaying 500 μl glucose (1 M) with 200 μl of the lipid mixture. The inner leaflet was prepared by overlaying 50 μl sucrose (1 M) with 200 μl of lipid mixture and mixed by shredding. The mixed inner leaflet content was gently pipetted on top of the outer leaflet phase. The tube was centrifuged at $2000 \times g$ for 3 min. GUVs were washed and resuspended in glucose (1 M). To check for PS in the inner leaflet, FITC-labeled recombinant lactadherin (rLactC2-FITC, 10 μg/ml) was mixed with sucrose and overlaid with the inner leaflet lipid mix, following the protocol above. To monitor the level of PS in the outer leaflet GUVs were incubated with rLactC2- FITC (10 μg/ml). GUVs were incubated rCPn0473 (20 μg/ml) in the presence of rLactC2 (10 μg/ml). PS translocation was determined via ImageJ analysis, by measuring the rLactC2-FITC signals associated with the GUVs. The median of the maximum intensity at three different positions on one GUV (up to 24 GUVs per experiment) was measured and compared to the background signal intensity. The LactC2-FITC signal intensity associated with GUVs in which PS was restricted to the inner leaflet was set to 1.

**Cellular PL externalization assays**. Externalized PS in HEp-2 cells was detected by incubation with 0.6 μl of annexin and 30 μl annexin staining buffer (0.1 M HEPES pH 7.4, 4.4 K NaCl 5 mM CaCl$_2$), or with 30 μl of the recombinant FITC- labeled, PS-binding domain of lactadherin in PBS (LactC2, 100 μg/ml) at 4 °C for 30 min prior to fixation with PFA. The phosphoinositides PI(4,5)P2 and PI(3)P were detected with 30 μl of the recombinant FITC-labeled PIP-binding domain of PLCγ and FYVE respectively. As a control, the PIP biosensors were incubated with PFA-fixed HEp-2 cells, which were permeabilized with saponin prior treatment to enable intracellular PIP-binding.

PL externalization after treatment with recombinant protein was analyzed as follows: Confluent HEp-2 cells were overlaid with 250 μl of culture medium containing 100 μg/ml recombinant protein. Cells were incubated with the protein solution for different times at 37 °C, and analyzed for PL externalization with the respective lipid marker as described above. The amount of externalized PS was quantified by measuring the total FITC using Nikon NIS software.

PS externalization during *Cpn* infection was analyzed as follows: Confluent HEp-2 cells were overlaid with 250 μl culture medium containing infectious EBs (MOI = 10). To measure the effects of EBs coated with recombinant protein on PS externalization, EBs were added to 100 μl of protein solution (1 μM) and incubated for 30 min at 4 °C prior to infection (MOI = 10). To measure the effects of selected antibodies on PS externalization, EBs were added to 100 μl of antiserum suspensions and incubated for 2 h at 4 °C. Infection was allowed to proceed up to 2 h at 37 °C. PS externalization was analyzed with either Annexin-V Fluor or LactC2

as described above. Only 4′,6-diamidino-2-phenylindole-stained and anti-LPS (*Cpn*) or anti-MOMP (*C. trachomatis*) positive chlamydial EBs were counted.

**Apoptosis assay**. To induce apoptosis in human epithelial cells, confluent HEp-2 cells were incubated in serum-free medium containing 2.5 μM staurosporine at 37 C for up to 16 h. To test for induction of apoptosis by rCPn0473, cells were incubated with 100 μg/ml Protein for up to 16 h. In order to analyze apoptosis early in infection, HEp-2 cells were infected with *Cpn* for 1 h at 37 °C. After extensive washing, cells were detached, pelleted (5 min at $300 \times g$) and resuspended in 100 μl HBSS. Induction of apoptosis was analyzed by Western blotting using caspase-3- and PARP-specific antibodies.

**Reporting summary**. Further information on research design is available in the Nature Research Reporting Summary linked to this article.

## Data availability
The data supporting the findings of this study are available within the paper and its extended data files. Data underlying Figs. 1a, b, d, e, 2a, b, d, 3b, c, e, 4a, c, d, g, Supp. Figs. 1a, b, d, 4e are provided as Source Data files. All other data are available from the corresponding author upon reasonable requests.

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

## Acknowledgements

We thank S. Birkelund (Aarhus University) and G. Zhong (University of Texas Health Science Center at San Antonio) for antibodies, J. Wiese (Heinrich-Heine-University Düsseldorf) for liposomes for pull-down experiments, S. Grinstein (University of Toronto) for the GST-Lactadherin-C2 expression vector, S. Wesselborg and C. Peters (Heinrich-Heine-University Düsseldorf) for the discussion of apoptosis and PS externalization, and for help with apoptosis-specific antibodies. We thank O. Kuge for the PS-deficient CHO cell line (PSA3). We are grateful to S. Hänsch (University of Düsseldorf) for providing us with the macro for microscopic analysis of the asymmetric GUVs with ImageJ software. We thank our reviewers for their constructive comments. J.N.G. was a scholarship holder funded by the Jürgen Manchot Foundation. T.F. was a scholarship holder of the Graduate School 'Molecules of Infection (MOI)' funded by the Jürgen Manchot Foundation. T.E. was supported by a Volkswagen Foundation grant (No. 92753) to Sven B. Gould. W.R. acknowledges support from the Excellence Initiative of the German Research Foundation (EXC 294), the Ministry of Science, Research and the Arts of Baden-Württemberg (Az: 33-7532.20), and a Starting Grant from the European Research Council (Programme: "Ideas", ERC-2011-StG 282105). This work was supported by the Deutsche Forschungsgemeinschaft, CRC 1208, project A05 to JHH.

## Author contributions

T.F. identified LIPP. T.F. and J.N.G. designed, conducted, and evaluated the liposome and cell culture experiments. T.E., J.N.G., and W.R. designed and conducted GUV experiments. J.N.G. and J.H.H. wrote the manuscript with input from all co-authors. J.H.H. initiated and led the project and discussed experiments.

## Conflict of interest

The authors declare no competing interests.
