## [Peer Review File · Nature Communications]

Reviewer #1 (Remarks to the Author):

Review on "A Chlamydia pneumoniae adhesin induces phosphatidylserine exposure at the host cell" by Galle et al.

Chlamydiae are important pathogens for humans or various animals. The species Chlamydia pneumoniae (Cpn) is causing respiratory infections in man. Additionally, it is discussed as an additional risk factor in the context of other diseases such as arteriosclerosis or Alzheimer. This genus of obligate intracellular bacteria possesses a unique productive cycle:

Infectious, but metabolically inert chlamydial elementary bodies (EBs) induce their uptake into host cells where they inhibit lysosomal fusion and remain in inclusions. There they develop to metabolically active and dividing reticulate bodies (RBs) until a new generation of infectious EBs is released after several days. Unfortunately only few chlamydial species or even strains such as C. trachomatis serovar L2 or C. muridarum can be modified genetically so far.

Previous work and background of the submitted manuscript:

Cpn0473 is a protein of Cpn which was first characterized in detail as described in a previous publication of Fechter et al. in Cellular Microbiology (2016) 18(8). Since the submitted manuscript deals with various new aspects concerning the interaction of Cpn0473 with the host cell after adhesion, it seems necessary for me to clarify what has been already published and is known based on the former article: First, the Outer membrane complex protein B (OmcB) binds to heparan sulfate moieties on the host cell surface. Then, the polymorphic membrane protein (Pmp) 21 adhesin and invasin interacts with the human epidermal growth factor receptor resulting in receptor activation, down-stream signalling and internalization of the bacteria. Blocking these two adhesion pathways did not completely abolish infection.

Based on that observation Cpn0437 was characterized as one additional surface factor involved in binding of Cpn EBs and invasion. CPn0473 is expressed late in the infection cycle. It is part of the infectious chlamydial cell surface. Soluble recombinant CPn0473 adheres to human epithelial HEp-2 cells.

The most important results of the previous publication have been the following: "Interestingly, in classical infection blocking experiments pretreatment of HEp-2 cells with rCPn0473 ... promotes dosedependently internalization by Cpn [and by C. trachomatis E but not by Ctr L2] suggesting an unusual mode of action for this adhesin ... [This] novel EB surface protein ... binds to cholesterol-rich regions in the bilayer of the host cell plasma membrane (PM) and contributes to internalization of Cpn via lipid rafts ... This ... promotion of infection by Cpn depends on two different domains within the protein ... The adhesion to human cells requires a 50 amino acids long stretch in the C-terminus (aa 307 to aa 356). The internalization enhancing effect is depending on the adhesion of CPn0473 to the human cell, but in addition requires the N-terminal 150 amino acids".

The three authors of the former paper who are also authors or coauthors of the submitted study announced then: "Thus, the CPn0473-induced boosting phenotype suggests an unusual mode of action for this adhesin. The molecular basis for this phenomenon is currently not understood but is the subject of a subsequent study."

Now, to the submitted manuscript, entitled: "A Chlamydia pneumoniae adhesin induces phosphatidylserine exposure at the host cell".

This study deals with the "identification of CPn0473 as a bacterial Phosphatidylserine (PS) translocase capable of specific and apoptosis-independent PS externalization during infection" uncovering "a novel mechanism by which intracellular pathogens enter host cells". Based on the functional characterization of this adhesin and the underlying mechanism of its unusual behavior the authors suggest "LIPP2 for 'Lipid-dependent Internalization Promoting Protein' as new name.

PS is a phospholipid with a negatively charged head-group. It is an essential but minor constituent of membranes from all eukaryotic cells, where it is unevenly distributed. In the healthy plasma membrane (PM) it is found preferentially in the inner leaflet. Various cytoplasmic proteins such as protein kinase C isoforms bind to it. In apoptosis and blood clotting disruption of this asymmetry occurs. For detections and characterization of PS PS-binding probes are used. Frequently, fluorescence-labelled annexin A5 (or annexin V), a non-covalently lipid-binding protein is used. It binds to all negatively charged lipids (when Ca^{2+} is present in relatively high amounts), PS being the most abundant. Much more specific for PS and Ca-independent is the binding of fluorescently labeled Lactadherin C2, often expressed as GFP-LactC2 (Kay J.G. et al. *Sensors* 2011, 11:1744-1755). Both methods are used in the submitted paper to detect PS in the inner or outer leaflet of the PM.

How convincing are the specific results and their interpretation?

In a first step, the authors screened for interaction partners of rCpn0473 detecting binding to nitrocellulose strips spotted with different lipids and confirming the orientating results with giant unilamellar vesicles (GUVs) made of defined mixtures of charged phospholipids (PLs). rCpn0473 did not bind to positively charged phosphatidylcholine (DOPC), and showed a markedly higher affinity for phosphatidylserine (DOPS) than for phosphatidic acid (DOPA, PA). It did not interact with Phosphatidylinositol 4,5-bisphosphate (PIP₂). The recognition of DOPS by rCpn0473 was dose-dependent. Supplementation with cholesterol increased binding to GUVs – in accordance to the binding of Cpn EBs to cholesterol-rich regions of the host PM. The residues 1-171 of rCpn0473 are essential for EB internalization. Their removal abrogated the interaction of the resulting mutant with DOPS GUVs, whereas fusion of aa1-176 permitted OmcB to adhere.

>> Thus, the N-terminus of Cpn0473 interacts with the PS.

Next, the PS-deficient mutant cell line CHO-PSA3 was used to assess the relevance of PS for infection: Infectivity was indeed decreased by 50% compared to regular PS-bearing CHO-K1 cells; and preincubation with rCpn0473 failed in the PSA3 mutant cell to increase EB internalization.

>> Thus, one can follow the conclusion of the authors that “Cpn0473 binds to PS, and that PS is essential for the rCpn0473-mediated boost ... of EB internalization and infectivity”.

PS is usually restricted to the inner leaflet of the PM. However, in apoptosis it switches to the outer leaflet - serving as a marker of this biological process. Incubation of HEp-2 cells with viable (but not heat-inactivated) EBs permitted within 5 to 60 minutes Annexin-binding. The association of adhering EBs with annexin-V signals was drastically lower for EBs of *C. trachomatis* serovar E and even lower for those of serovar LGV. Experiments with the more specific rLactC2 confirmed (part of) this results. Cpn and rLactC2 signals accumulated in cholesterol-enriched (cholera toxin-positive) membrane domains.

>> These results are in accordance to the presumed role of PS, located on the surface of host cells, in Cpn EB adherence and infection.

In addition, rCpn0473 led to PS externalization (at the binding site of LIPP) in a time and dose dependent manner, visualized by Annexin V and rLactC2-FITC. In contrast, rPMP21 or rOmcB had no such effect. The rCpn0473 dose- and time-dependent externalization of PS in the PM was sensitive to cholesterol depletion (by methyl-beta-cyclodextrin; M β CD). Intriguingly, no cleavage of Caspase 3 occurred in this case.

>> These results indicate that rCpn0473 leads, in the absence of apoptosis, to the externalization of PS in the PM.

As mentioned in my introduction of this review, Cpn can presently not be modified genetically. Therefore, targeted mutants of Cpn which do not express Cpn0473 or which overexpress this adhesin cannot be generated and functionally studied.

To overcome this technical limitation and to simulate as close as possible such functional gain and loss experiments, the authors preincubated and "coated" EBs with rCpn0473 or they used antibodies blocking either this adhesin or, as a negative control, Pmp21. EBs with additional rCpn0473 caused a significant increase in EB-associated PS signals, and blocking with (only) the anti-Cpn0473 antibody led to a decrease of this signal.

>> Thus, the amount of free Cpn0473 on the surface of Cpn EBs strongly influenced the interaction with and externalization of PS.

Using mutants of rCpn0473 the authors could functionally map this protein.

>> There results show that the N-terminal aa 1-149 are essential for PS externalization.

A biotin-protection experiment against (extracellular) exposure of host cells with β -mercaptoethanol using two different His-tagged rCpn0473 derivates which contained a biotinylated single cysteine either at the N- or the C-terminus ...

>> ... suggests that the C-terminus of Cpn0473 (with a predicted transmembrane region "TM" at aa 396 - 425) might be inserted into the host PM.

Next, DOPS-GUVs were loaded with carboxyfluorescein (CF; MW > 350 kD) and incubated in the presence of Alexa594-labeled rCPn0473 (55 kD). Upon incubation with rCPn0473, the CF signal within the GUV lumen decreased within a few minutes followed by an increase in the luminal Alexa594 signal. A TM deletion variant (Δ TM) had no such effects.

>> One can follow the conclusions that binding of rCPn0473 makes the membrane of DOPS-GUVs permeable to small and large molecules. However, the overall stability of the GUVs seems not to be affected.

To demonstrate directly the "PS translocator activity" of rCPn0473, GUVs with asymmetric distribution of PS were prepared. In their native form they bear PS at their inner leaflet. They were incubated for up to 30 minutes with the adhesin. That led immediately to a switch of the PS to the outer leaflet as demonstrated by binding of FITC-rLactC2.

>> Thus, indeed, a rapid PS translocator activity" of rCPn0473 could be demonstrated by this experiment.

>>>>

Taken together, the experiments are sound and convincing, and their interpretation is highly comprehensible. All results are in accordance with the suggested role of rCpn0473, or LIPP, as an ATP and Ca²⁺ independent PS translocator which is located at the surface of EBs of Cpn and which participates (in cooperation with OmcB and Pmp21) first in adhesion to the host cell and which then leads to exposure of PS in the outer leaflet of the host PM close to the EBs. rCpn0473 or LIPP acts without causing apoptosis, somehow inducing the uptake of the EBs.

To my knowledge, as claimed by the authors of this study, LIPP might be indeed the first known lipid translocator protein that can gain access to the inner leaflet of the PM from the extracellular side.

As mentioned by the authors, it still remains unclear how chlamydia induce this translocation, i.e. which host signal molecule is activated to cause this effect in the PM. And it is also so far only speculation, although a rational one, that PS translocation near the adherent Cpn EBs might influence the outward curvature at the edges of the invaginating PM.

Nevertheless, these results go far beyond the first characterization of Cpn0473 in the former publication. The findings of this elegant study are new and highly interesting - characterizing in

detail the mechanism triggered by a chlamydial adhesin and resulting in its surprising additional function, the boost of internalization and infection. Probably, other intracellular pathogens might use similar mechanisms. Thus, we learn from this study most likely more about the host cell biology and the involved eukaryotic machinery for the uptake of pathogens in general. Therefore, these findings are not only of interest to others in the community but also to a wider field.

The statistical analysis seems to be valid. Given the level of technical detail provided, other researchers should be able to reproduce the work.

Minor points:

"LIPP" is explained in line 135 "... protein hereafter as LIPP (for 'Lipid-dependent Internalization Promoting Protein')..." - However this abbreviation is already used several times from line 77 on "Indeed, we observed PS externalization at the binding site of LIPP in a time and dose dependent manner..."

Reviewer #2 (Remarks to the Author):

This manuscript describes the Chlamydia adhesin CPn0473 as a novel phosphatidylserine (PS) translocase. Using both in vitro and in vivo assays based on liposomes and Hep-2 cells, respectively, the authors make a case for this protein as PS translocase that triggers PS exposure at the host plasma membrane. In short, they claim to have uncovered a novel mechanism by which intracellular pathogens enter host cells. The authors carefully map the domains responsible for membrane binding and PS externalization, show that the protein induces pore formation, and demonstrate that the C-terminal end of CPn0473 inserts into the cell membrane. While of clear interest to the field, technically sound and worthwhile to publish, the results presented in this study fall short of adequately supporting the major conclusion of the authors. Key issues with the manuscript are delineated below:

1. The authors suggest that the adhesin CPn0473 is a novel PS-specific translocase but have not tested other lipids in their elegant assays based on asymmetric giant vesicles to rule out a general lipid scrambling effect. In fact, pore-forming proteins are known to induce scrambling of all lipid species during membrane insertion. Thus, the easiest explanation for the results observed would be a non-specific scramblase function for CPn0473. As such, this would be still novel and interesting as the authors demonstrate that other adhesins do not exert this effect on cells. Exposure/redistribution of other lipids, e.g. phosphatidylethanolamine or PIPs, should be tested on cells and liposomes.
2. The name PS translocase implies a unidirectional transport mechanism for PS. This has not been tested. In fact, the authors need to exclude a scrambling mechanism and need to show specificity to justify their conclusion.
3. The authors suggest that the PS externalization triggered by the adhesin CPn0473 is independent on calcium; this conclusion seems to be based on in vitro experiments using giant liposomes in calcium free-buffer. However, given the pore-forming activity of CPn0473, PS exposure in cells might be simple triggered by calcium entry from the medium, thereby activating cellular scramblases at the plasma membrane. This needs to be ruled out. In fact, it is not clear whether the overlay and liposome binding assays were performed in the absence or presence of calcium and how these conditions compare to the cell infection assays.
4. The PS externalization triggered by CPn0473 is quite slow in both in vitro and in vivo. This is unexpected for a translocase/scramblase which facilitates rapid lipid transbilayer movement. The authors should discuss this point.

Specific comments:

- The authors refer to flippases, floppases and scramblases as proteins establishing and maintaining phospholipid asymmetry. This is not correct; transbilayer lipid asymmetry is established and maintained by ATP-driven phospholipid flippases/floppases but counteracted by scramblases that operate without ATP. There are excellent reviews in the field that can be cited.
- Phosphatidylcholine is not a positively charged lipid.
- The authors suggest that cholesterol is enhancing binding of full-length CPn0473 based on single vesicle analysis using giant liposomes. Preparations of giant vesicles are known for their heterogeneity and individual vesicles will differ in their lipid composition. In fact, even the presence of cholesterol and the respective phospholipid at the expected levels in the vesicles should be verified to make this conclusion.
- Figure 2c does not allow the reader to identify the plasma membrane.
- In figures presenting normalized data, the value used for normalization should be provided.
- The incubation time on ice during the adhesion assay, the procedure for cell detachment and the buffer composition of HBBS (with/without Ca²⁺, Mg²⁺) is missing (line 340, 341).
- Preparation of PS liposomes (line 361) and Sigma-lipid liposomes (line 396) is not explained; where SUVs or LUVs prepared in both cases?
- Buffer composition of PBS (with/without Ca²⁺, Mg²⁺) is not given (line 390ff).

Replies to reviewers' comments.

Reviewer 1:

- 1) Taken together, the experiments are sound and convincing, and their interpretation is highly comprehensible. All results are in accordance with the suggested role of rCpn0473, or LIPP, as an ATP and Ca²⁺ independent PS translocator which is located at the surface of EBs of Cpn and which participates (in cooperation with OmcB and Pmp21) first in adhesion to the host cell and which then leads to exposure of PS in the outer leaflet of the host PM close to the EBs. rCpn0473 or LIPP acts without causing apoptosis, somehow inducing the uptake of the EBs.

We thank the reviewer for the enthusiastic support.

- 2) "LIPP" is explained in line 135 "... protein hereafter as LIPP (for 'Lipid-dependent Internalization Promoting Protein')..." - However this abbreviation is already used several times from line 77 on "Indeed, we observed PS externalization at the binding site of LIPP in a time and dose dependent manner..."

The reviewer is right. We now refer to the protein as CPn0473 in lines 78 and 91, and introduce the term 'LIPP' on (what is now) line 146.

Reviewer 2:

- 1) The authors suggest that the adhesion CPn0473 is a novel PS-specific translocase but have not tested other lipids in their elegant assays based on asymmetric giant vesicles to rule out a general lipid scrambling effect. In fact, pore-forming proteins are known to induce scrambling of all lipid species during membrane insertion. Thus, the easiest explanation for the results observed would be a non-specific scramblase function for CPn0473. As such, this would be still novel and interesting as the authors demonstrate that other adhesins do not exert this effect on cells. Exposure/redistribution of other lipids, e.g. phosphatidylethanolamine or PIPs, should be tested on cells and liposomes.

The reviewer is correct in stating that, in the original MS, we only tested the phospholipid PS in our translocation assays - because CPn0473 showed the highest affinity for PS-containing synthetic membranes (Fig. 1b). However, because a phospholipid-specific translocase activity of CPn0473 is a very intriguing possibility, we decided to additionally test the negatively charged phosphoinositides PI(4,5)P₂ and PI(3)P, both of which are normally confined to the inner (cytoplasmic) plasma membrane leaflet. To our knowledge, no PIP-specific probes are commercially available. Therefore, Dominik Spona in our lab generated two recombinant fusion proteins: (i) the PI(4,5)P₂-binding domain from PLC γ ¹, and (ii) the PI(3)P-binding domain of FYVE¹, both of which were C-terminally tagged with GST (Spona and Hegemann, unpubl.). Fluorescein-labelled rPLC γ and rFYVE bound PI(4,5)P₂- and PI(3)P-containing GUVs respectively (Supplementary Fig. S1e and f). Moreover, rPLC γ showed binding to PI(4,5)P₂ on membrane lipid strips (Supplementary Fig. S1a).

While the PI(4,5)P₂ and PI(3)P biosensors detected intracellular PI(4,5)P₂ and PI(3)P in human cells permeabilized with MeOH, no fluorescence was observed in non-permeabilized cells, confirming the intracellular nature of both PIP species (Supplementary Fig. 2c). Interestingly, rCPn0473-treated HEp-2 cells also showed no signals for both biosensors, indicating that exposure of human cells to rCPn0473 does not induce translocation of the phosphoinositides PI(4,5)P₂ and PI(3)P to the outer membrane leaflet (Supplementary Fig. 2c). However, we readily observed externalization of phosphatidylserine upon rCPn0473 treatment (Supplement Fig. 2b). Because we saw

no translocation of PI(4,5)P₂ and PI(3)P on cells treated with rCPn0473, we did not perform these experiments on GUVs.

In the revised MS we have added the following statement (lines 80-86): “Importantly, when using the FITC labeled PI(4,5)P₂-binding domain of PLC γ and the PI(3)P-binding FYVE domain of the Hgs (Supplementary Fig. 1a,e,f), we did not detect any externalization of phosphatidylinositol 4,5-bisphosphate (PI(4,5)P₂) and phosphatidylinositol 3-phosphate (PI(3)P), two other negatively charged phospholipids that are normally restricted to the inner leaflet of the PM, upon the addition of rCPn0473 to HEP-2 cells (Supplementary Fig. 2c), evidencing specificity of rCPn0473 for PS.”

- 2) *The name PS translocase implies a unidirectional transport mechanism for PS. This has not been tested. In fact, the authors need to exclude a scrambling mechanism and need to show specificity to justify their conclusion.*

We agree with the reviewer that we had not tested directionality of PS translocation. The question about the phospholipid specificity of CPn0473 has been answered in point 1 (see above).

The experiments in the original MS show that incubation with rCPn0473 mediates PS translocation from the inner to the outer leaflet of GUVs harboring PS in the inner leaflet (Fig. 4f,g). In order to test the directionality of CPn0473-mediated PS transport, we have performed additional experiments with asymmetric GUVs. In the new data set (Suppl. Fig. 4d,e), rLactC2 as a marker for PS was prepacked into the GUV lumen, while PS was loaded into the outer leaflet of the GUV membrane (PS_{out}-GUV). We then tested whether exposure of these GUVs to rCPn0473 would result in translocation of PS from the outer to the inner leaflet where it would recruit the PS marker rLactC2 to the inner membrane leaflet. We observed no such rLactC2 binding to the inner leaflet of these GUVs in the presence of rCPn0473. This finding is compatible with a (directed) translocase activity of LIPP. These results demonstrate that binding of rCPn0473 to a low-PS membrane environment rapidly mediates directional PS translocation from the high-PS (inner) to the low-PS (outer) leaflet in a synthetic membrane system. This finding nicely resembles the initial cellular stage, where rCPn0473 triggers PS externalization upon binding to the extracellular, low-PS plasma membrane leaflet

We have added the following statement to the revised MS (lines 139-143):

“On the other hand, when PS_{out}-GUVs, prepacked with FITC-rLactC2, were exposed to external rCPn0473, we observed no FITC-rLact2 fluorescence at the inner leaflet (Supplementary Fig. 4d,e) (Supplementary Fig. 4d,e). These results argue against a scrambling mechanism for rCPn0473, and thus support directional rCPn0473-mediated PS translocation from the inner (PS-high) to the outer (PS-low) leaflet in a synthetic membrane system.”

- 3) *The authors suggest that the PS externalization triggered by the adhesin CPn0473 is independent on calcium; this conclusion seems to be based on in vitro experiments using giant liposomes in calcium free-buffer. However, given the pore-forming activity of CPn0473, PS exposure in cells might be simple triggered by calcium entry from the medium, thereby activating cellular scramblases at the plasma membrane. This needs to be ruled out. In fact, it is not clear whether the overlay and liposome binding assays were performed in the absence or presence of calcium and how these conditions compare to the cell infection assays.*

We agree with the reviewer that our manuscript was not entirely clear in this respect. In fact, all synthetic membrane assays (liposome assays, including liposome pulldown, GUV binding and lipid-flipping assays at asymmetric GUVs) were performed in calcium-free

buffer. Thus, the externalization of PS by our protein is independent of calcium. All cell-based experiments were performed in DMEM media containing 1.8 mMol Ca²⁺.

Moreover, we have now tested the effect of different extracellular calcium levels on rCPn0473-induced PS externalization in HEp-2 cells (Fig 3b). We observed no significant alteration in the level of PS externalization when cells were treated with rCPn0473 in the presence of elevated calcium levels.

We now have added the following statement in the revised manuscript (lines 88-91):

“Since calcium entry might trigger cellular PS exposure, we tested different calcium concentration in the medium of Hep-2 cells. Notably, the level of rCPn0473-induced PS externalization is not enhanced by elevated extracellular calcium levels (Fig 3b).”

- 4) *The PS externalization triggered by CPn0473 is quite slow in both in vitro and in vivo. This is unexpected for a translocase/scramblase which facilitates rapid lipid trans bilayer movement. The authors should discuss this point.*

Translocases and scramblases are integral membrane proteins located in the plasma membrane (PM), and are activated by ATP, Ca²⁺ and proteolytic cleavage. In contrast, LIPP is an adhesin located at the cell surface of the infectious chlamydial EBs, which only begins to interact with the PM once the EB makes contact to the host cell, probably via the chlamydial adhesin OmcB and host cell glycosaminoglycans. It is very likely that the CPn0473 – PM interaction is a complex process, possibly requiring conformational changes and oligomerization of CPn0473 either before or during membrane insertion. This process may affect lipid trans bilayer movement. However, we believe that any further discussion of issues such as translocation rate would be premature in the absence of further insight into the CPn0473-mediated translocation process itself.

Accordingly, we have revised the text as follows (lines 147-153):

“... Our model implies that LIPP interacts via its BD domain with the PM via an as yet unknown cell-surface structure associated with cholesterol-rich microdomains. The LIPP - PM interaction likely triggers both conformational changes within the protein and oligomerization. Together, these changes enable the N-terminal domain of LIPP to insert into the PM and translocate PS from the inner to the outer leaflet of the bilayer – which in turn enhances EB internalization. This scenario could explain why the LIPP translocation complex acts rather slowly by comparison with known integral membrane flippases and scramblases.”

- 5) *The authors refer to flippases, floppases and scramblases as proteins establishing and maintaining phospholipid asymmetry. This is not correct; transbilayer lipid asymmetry is established and maintained by ATP-driven phospholipid flippases/floppases but counteracted by scramblases that operate without ATP. There are excellent reviews in the field that can be cited.*

The reviewer is right.

We therefore rephrased the first sentence of the main text (lines 27-31):

“The mammalian plasma membrane (PM) exhibits a distinct phospholipid asymmetry which is established and maintained by specific ATP-driven lipid translocators (flippases, floppases)^{1,2}. Movement of phosphatidylserine (PS) from the inner to the outer leaflet of the PM is called externalization, a process executed by scramblases, which occurs in apoptotic cells (triggering their removal by phagocytes) and during platelet activation^{3,4}”.

- 6) *Phosphatidylcholine is not a positively charged lipid.*

The Reviewer is correct.

We have changed the statement (line 45) to
“... consisting of phosphatidylcholine (DOPC).”

- 7) *The authors suggest that cholesterol is enhancing binding of full-length CPn0473 based on single vesicle analysis using giant liposomes. Preparations of giant vesicles are known for their heterogeneity and individual vesicles will differ in their lipid composition. In fact, even the presence of cholesterol and the respective phospholipid at the expected levels in the vesicles should be verified to make this conclusion.*

We agree with the reviewer that there is a certain level of heterogeneity in lipid composition between individual GUVs within a GUV population. For this very reason, we believe that determining the actual amount of a lipid species in single GUVs would only confirm this heterogeneity. In fact, we already take this compositional variation into account by displaying binding efficiencies as mean values with their corresponding error bars of more than 100 different GUVs. Moreover, the cited suggestion is based on comparison of GUVs made of cholesterol-containing lipid mixtures with GUVs that were completely free of cholesterol, for which the issue does not arise.

- 8) *Figure 2c does not allow the reader to identify the plasma membrane.*

In order to visualize the plasma membrane, we have performed additional experiments. *C. pneumoniae*-infected cells were stained with an anti-EGFR-specific antibody to decorate the plasma membrane of the host cell, while externalized PS was detected via annexin-V-FLUOS (Fig 2c). The human EGFR is recruited by *C. pn.* early in infection.²

We therefore rephrased the figure legend of Fig. 2c (lines 198-199):

“Externalized PS was stained with annexin-V-FLUOS prior to fixation, followed by staining with DAPI and an anti-EGFR antibody.”

- 9) *In figures presenting normalized data, the value used for normalization should be provided.*

We provide a Source Data File containing the raw data for all graphs and tables as a separate file together with the revised submission.

- 10) *The incubation time on ice during the adhesion assay, the procedure for cell detachment and the buffer composition of HBBS (with/without Ca²⁺, Mg²⁺) is missing (line 340, 341).*

We thank the reviewer for this comment.

We have added the information requested by the reviewer in the revised manuscript.
“... soluble recombinant protein of interest (100 µg/ml) at 4 °C on ice for 15, 30 or 60 min...” (lines 370-371)

“... After extensive washing, cells were detached, using Cell Dissociation Solution, pelleted (5 min at 300xg) and resuspended in HBSS (100 µl)...” (line 384-386)

“HBSS buffer (Thermo Scientific, #14175-053) used in this work consists of 5.3 mM Potassium Chloride (KCl), 0.4 mM Potassium Phosphate monobasic (KH₂PO₄), 4.2 mM sodium bicarbonate (NaHCO₃), 137.9 mM sodium chloride (NaCl), 0.3 mM sodium phosphate dibasic (Na₂HPO₄), 5.6 mM D-Glucose (Dextrose).” (lines 325-328)

- 11) *Preparation of PS liposomes (line 361) and Sigma-lipid liposomes (line 396) is not explained; where SUVs or LUVs prepared in both cases?*

We thank the reviewer for this comment.

We have added the information requested by the reviewer in the revised manuscript. In both cases, liposomes with an expected diameter of less than 100nm, so called SUVs, were prepared. All liposomes were prepared via the *freeze-thaw-sonification*-method, which we now have explained in more detail in the ‘materials and methods’ section. The modified text now reads (lines 428-432):

“... were prepared via the freeze-thaw-sonification method (generous gift from J. Wiese, Heinrich-Heine-University Düsseldorf). Lipids were resuspended in chloroform and

acetone and stirred for 2 h at RT. After precipitation of the phospholipids overnight at 4 °C, lipids were dissolved in diethyl ether and dried under nitrogen. The phospholipid-pellets were dissolved in HBSS and sonicated in a water bath (30 x 5 seconds).”

12) *Buffer composition of PBS (with/without Ca²⁺, Mg²⁺) is not given (line 390ff).*

The PBS buffer used neither contains Ca²⁺ nor Mg²⁺.

We have added the buffer composition in the revised manuscript in the first paragraph of the ‘Materials and Methods’ section (lines 324-328):

“... . PBS buffer used in the experiments consists of 137 mM NaCl, 2.7 mM KCl, 10 mM Na₂HPO₄, 1,8 mM KH₂PO₄. ...”

References

- 1 Balla, T. & Varnai, P. Visualization of cellular phosphoinositide pools with GFP-fused protein-domains. *Curr Protoc Cell Biol* **Chapter 24**, Unit 24 24, doi:10.1002/0471143030.cb2404s42 (2009).
- 2 Molleken, K., Becker, E. & Hegemann, J. H. The Chlamydia pneumoniae invasin protein Pmp21 recruits the EGF receptor for host cell entry. *PLoS Pathog* **9**, e1003325, doi:10.1371/journal.ppat.1003325 (2013).

Reviewers' comments:

Reviewer #2 (Remarks to the Author):

The authors have largely addressed my concerns. Their manuscript contains a great amount of work.

However, the issue of lipid scrambling could be clarified more as outlined below.

(1)The authors prepared two PIP sensors and show that these sensors detect PIP lipid after methanol permeabilization. This technique is problematic as it involves the use of an organic solvent. The organic solvents dissolve lipids from cell membranes making them permeable to proteins, e.g. antibodies. This brings up the question what the two PIP sensors are detecting after methanol permeabilization. Furthermore, the precise conditions of the methanol permeabilization are not described (at which temperature? At which methanol concentration? How long?). This issue should be clarified.

(2)The authors have now tested in their elegant GUV assay the directionality of the PS transport. For this, they loaded rLactC2 as a marker for PS into the GUV lumen. While this is a great approach, it is not clearly explained how they have solved the permeabilization effect of CPn0473 as shown in Figure 4c-f in the manuscript. Should rLactC2 not leak out of the GUVs after CPn0473 treatment as under the same conditions it can leak in?

(3)The authors have now also measured the effect of different extracellular calcium concentrations at the millimolar range on CPn0473-induced lipid scrambling. However, calcium-activated lipid scrambling is sensitive to the presence of calcium concentrations at the micromolar range. Thus, a control is missing on cells incubated in calcium free medium supplemented with EGTA which completely eliminates all extracellular calcium. Given the amount of data in this manuscript, it seems unfair to ask the authors to perform this experiment at this point, but they should at least address this question more carefully in the text.

(4)The title of Supplementary figure 3 implies that intracellular calcium levels have been measured but such data are not presented.

Replies to reviewers' comments.

Reviewer 2:

The authors have largely addressed my concerns. Their manuscript contains a great amount of work.

(1) *However, the issue of lipid scrambling could be clarified more as outlined below. The authors prepared two PIP sensors and show that these sensors detect PIP lipid after methanol permeabilization. This technique is problematic as it involves the use of an organic solvent. The organic solvents dissolve lipids from cell membranes making them permeable to proteins, e.g. antibodies. This brings up the question what the two PIP sensors are detecting after methanol permeabilization. Furthermore, the precise conditions of the methanol permeabilization are not described (at which temperature? At which methanol concentration? How long?). This issue should be clarified.*

Initially, the reviewer had asked whether rCPn0473 has scrambling activity, i.e. translocates lipids other than PS. To answer this question, we generated two recombinant PIP lipid biosensors and showed that each recognizes its specific lipid target (PI(3)P or PI(4,5)P₂) in artificial membranes (giant luminal vesicles, GUV) (Fig. S1 e, f). Next we showed that incubation of *living* human cells with rCPn0473 does not lead to the flipping of PI(3)P or PI(4,5)P₂ – both of which are located in the cytoplasmic leaflet of the plasma membrane (Fig. S2c) – into the outer leaflet. Together, these findings corroborate our contention that rCPn0473 specifically interacts with the lipid phosphatidylserine. As an additional control, we showed that the biosensors are able to bind *intracellular* PI(3)P and PI(4,5)P₂ structures only if the human cells have been fixed and permeabilized with methanol [by incubation with cold (-20°C) methanol for 5 min at room temperature].

The reviewer is correct in stating that methanol treatment can to some extent mobilize small molecules like lipids during fixation. Importantly, this does not affect the significance of the rCPn0473 experiment, as in this case the living human cells are neither fixed nor permeabilized, but simply incubated first with recombinant rCPn0473 and then with the recombinant biosensor.

However, as an additional control, we tested our PIP biosensors on human cells that had been subjected to a different permeabilization and fixation protocol: After PFA fixation cells were permeabilized with saponin, which is a mild detergent and known to be less harmful to membrane components like lipids. Here again, we observed that both biosensors detected PIP staining of intracellular structures in PFA/saponin-treated human cells. We have incorporated these new controls into Fig. S2c in place of the methanol controls. As seen before in our MeOH-permeabilized cells, FITC-labelled-rPLC γ binds to the host plasma membrane, while FITC-labelled-rFYVE binds to vesicle-like compartments, probably early endosomes.

In the manuscript, we have altered the figure legend for Fig. S2c to read as follows: “Human HEP-2 cells were treated with or without rCPn0473 (100 μ g/ml, 60 min at 37 °C) in DMEM medium. PI(4,5)P₂ was detected by rPLC γ , and PI(3)P was detected by rFYVE prior to fixation with PFA. As a control, cells were fixed with PFA (10 min, 4 °C) and permeabilized with 2 % saponin (30 min, 4 °C) prior to incubation with the respective lipid biosensor, to ensure that the latter had access to intracellular PIP.” (lines 275 - 279)

Furthermore, we have added the following sentence in the Methods section to explain our permeabilization and staining protocol:

“Where necessary, cells were permeabilized with 2 % saponin prior to staining with an antibody or marker protein. In this case, 0.5 % saponin was present during the whole staining process.” (lines 369 – 370)

- (2) *The authors have now tested in their elegant GUV assay the directionality of the PS transport. For this, they loaded rLactC2 as a marker for PS into the GUV lumen. While this is a great approach, it is not clearly explained how they have solved the permeabilization effect of CPn0473 as shown in Figure 4c-f in the manuscript. Should rLactC2 not leak out of the GUVs after CPn0473 treatment as under the same conditions it can leak in?*

The reviewer is right in that rCPn0473 binding to GUVs induces permeabilization of the GUV membrane. Importantly however, the permeabilization effect is a time-dependent process, as shown in the kinetics experiment in Fig. 4c+d. Our measurements revealed that large molecules like rCPn0473 (55kDa) and small molecules like carboxyfluorescein (CF, 370 Da) cross the membrane barrier at different rates. A 50% loss of preloaded CF is observed within approximately 2.5 min, while leakage of 50% of the maximal Alexa594-CPn0473 uptake from the GUV lumen takes more than three times as long (8.5 min).

Importantly, detection of translocated PS on asymmetric GUVs by rLactC2 (45 kDa) occurs immediately (≤ 10 sec) after addition of rCPn0473 to the GUV vesicles (Fig. 4f+g). At this point (termed ‘0 min’ in Fig. 4g), rCPn0473-induced permeabilization is minimal (as seen in Fig. 4c+d). Thus, we conclude that there is minimal leakage of rLactC2 into the lumen of the rCPn0473-treated GUVs at this time. However, leakage of rLactC2 (45 kDa) across the membrane might occur at later time points.

- (3) *The authors have now also measured the effect of different extracellular calcium concentrations at the millimolar range on CPn0473-induced lipid scrambling. However, calcium-activated lipid scrambling is sensitive to the presence of calcium concentrations at the micromolar range. Thus, a control is missing on cells incubated in calcium free medium supplemented with EGTA which completely eliminates all extracellular calcium. Given the amount of data in this manuscript, it seems unfair to ask the authors to perform this experiment at this point, but they should at least address this question more carefully in the text.*

We agree with the reviewer that it would be very interesting to investigate PS translocation in the absence of calcium ions. Unfortunately, we cannot make use of EGTA/EDTA in our experimental setup. We have investigated PS translocation by rCPn0473 on adherent epithelial cells and observed that EDTA/EGTA leads to the dissociation of adherent cells, which makes it impossible to study the PS translocation effect of rLIPP by microscopy.

Therefore, in a new approach, we replaced the calcium-containing DMEM medium with calcium-free HBSS buffer. Preliminary results suggest that PS translocation by rCPn0473 is not affected by the absence of extracellular calcium - although the cells look much more stressed in HBSS (see micrographs below):

Thus, in future we plan additional experiments to address this question in more detail. We followed the reviewer's advice and have modified our manuscript to address this point more carefully in our MS:

"..., although a potentially beneficial effect of intracellular ATP- or Ca²⁺-stores cannot be ruled out at the moment." (lines 154-155)

(4) *The title of Supplementary figure 3 implies that intracellular calcium levels have been measured but such data are not presented.*

The reviewer is right. We have changed the title to "CPn0473-induced PS externalization is not affected by disruption of the host cytoskeleton" (lines 281-282)

Reviewer #2 (Remarks to the Author):

The authors have adequately address all points of concerns and revised the manuscript.
Congratulation to this nice piece of work.